# Study on the Applicability of Dynamic Factor Standards by Comparison of Spring Constant Based Dynamic Factor of Ballasted and Concrete Track Structures

**Jaeik Lee [1,2], Kyuhwan Oh [2], Yonggul Park [2] and Junhyeok Choi [3,\*]** 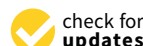

[1]   Department of Railway Engineering, Seoul Nation University of Science and Technology, Seoul 01811, Korea; jaeik2@illinois.edu

[2]   Department of Civil and Environmental Engineering, University of Illinois at Urbana-Champaign, Champaign, IL 61820, USA; Kyuhwan.Oh@seoultech.ac.kr (K.O.); ygpark@seoultech.ac.kr (Y.P.)

[3]   Department of Civil Engineering, Bucheon University, Bucheon 14642, Korea

\*   Correspondence: cjunh@bc.ac.kr

**Abstract:** Dynamic factor evaluation method calculation methods outlined by Eisenmann ($DAF_{Eisenmann}$) and the American Railway Engineering Association ($DAF_{Area}$) are used to calculate the dynamic factor during design and for trackside measurement, respectively, in nations where the construction of concrete track structures is relatively new. In this situation, dynamic factor calculation methods may be incorrect, and this is demonstrated by comparison of the respective track types' total spring constant. A finite element analysis of a standard design railway track is conducted, and the design total spring constant (TSC, or K) obtained from the time history function analysis is compared to the TSC of existing tracks through trackside measurement results. The comparison result shows that TSC obtained by finite element analysis result is 22% higher than that of the trackside measurement value, indicating that the TSC is conservative in the current track design. Considering the proportional relationship between TSC and dynamic factor, it is estimated that the dynamic factor currently being applied in track design is also conservative. Based on these findings, an assessment of the applicability of different dynamic factors ($DAF_{Eisenmann}$ and $DAF_{Area}$), theoretical calculation and field measurement ($DAF_{Field}$) using the probabilistic analysis of wheel loads from the field measurement data is conducted. A correlative analysis between $DAF_{Eisenmann}$ and $DAF_{Area}$ shows that $DAF_{Eisenmann}$ and $DAF_{Area}$ were estimated to be higher by 33% and 27% in ballasted track and by 39% and 30% in concrete track than the dynamic factor derived from field measurement, respectively, which indicates that the dynamic factor currently in use can potentially lead to over-estimation in track design and maintenance.

**Keywords:** dynamic amplification factor; total spring constant of track; field measurement; finite element method; railway system management; railway maintenance

## 1. Introduction

The railway system is intended to provide a fast, comfortable and safe public transportation, and proper design, maintenance and evaluation of track stability performance is essential to avoid derailment or other train-related accidents due to rail defects. However, in developing countries or countries that reference other international standards for design and maintenance of tracks, track design factors are not necessarily communal, and this is currently the case with regards to dynamic factor calculation methods. Dynamic factor is derived by the ratio of the dynamic wheel load to the static



wheel load—where dynamic wheel load is larger than static wheel load owing to the rail surface roughness, track irregularity, train speed and track support stiffness—and dynamic factor is a key factor for dynamic stability performance evaluation in railway track structures [1].

It is common knowledge that there are various dynamic factor calculation methods existing in the academic field of railway engineering, and each method employs different methods. It is difficult, with the present state of research results, to claim whether one method is better than the other. Despite these circumstances, it is important to consider the risks of applying the same model of dynamic factor calculation on tracks structures with different stiffness properties. In countries such as Korea, China, and parts of South East Asia and the Middle East, a high-speed railway system is either introduced all together, or new, concrete type track structures are being constructed [2]. With the employment of new technologies and structures, it should follow that the design specifications and dynamic performance evaluation methods should cater to these changes. However, these nations are adopting conventional standards from overseas, in most cases adopting well-known standards such as the United Railway Code (UIC) or American Railway Maintenance-of-Way Association (AREMA) codes [3].

While the codes themselves are not incorrect, in the process of adopting the codes, there is a significant lack in the consideration of the difference in the environmental conditions, as well as the materials and track properties that are unique to the national circumstances. In consideration of this, Korea for example has undergone a change in the dynamic factor evaluation calculation method in their national standard [4]. Originally, dynamic factor calculation of railway tracks consisted of employing only the speed of the vehicle as a variable, based on the wheel load calculation derived from Zimmermann's method. However, since the development of high-speed railway, another calculation method based on Eisenmann's equation began to be used [5]. In this transition period, certified track dynamic stability evaluation companies and institutions are still using both calculation methods, without properly discerning the difference between the two methods.

It is important to investigate what the potential consequences of using the different codes could be in terms of dynamic performance evaluation of railway track structures by actually conducting two separate evaluation methods of the dynamic properties of the track. As a demonstration of this proposal, this study conducts an experimental evaluation of concrete and ballasted track structures, using two separate design specification dynamic factor calculation methods (theoretical dynamic factors). One method employs Eisenmann's formula (hereby referred to as $DAF_{Eisenmann}$, where the required condition is the track quality and speed of the vehicle, and one method based on the dynamic factor calculation method outlined in the Korean National Railway Code (hereby referred to as the $DAF_{Area}$). Dynamic factors calculated using these theoretical methods is compared to the dynamic factors obtained from trackside measurement (hereby referred to as $DAF_{Field}$), calculated using a formula based on the standard deviation of wheel load fluctuation. The comparison serves as a basis for an investigation of the relation between the dynamic factors of the respective track types (ballasted and concrete tracks) to assess the applicability of the different dynamic factor calculation standards.

## 2. Theoretical Discussion

### 2.1. Dynamic Factor Considering the Stiffness or Modulus of the Track

Dynamic factor is a parameter for track design and dynamic stability evaluation for maintenance and track design [6]. The principle idea of the dynamic factor is to understand, or evaluate, the dynamics of the railway track structure, where the load caused or applied by the moving train on the track is compared to the load expected by the load capacity of the track (represented by the static load of the vehicle design specification) in a form of a coefficient. Research trends related to this subject are as follows. Researchers in University of Illinois Urbana-Champaign shows various ways of calculating dynamic factors in all over the world [7]. Also, they have conducted the calculation to derive the dynamic factor depending on the calculation method they mentioned and improved the difference of values they calculated. Refer to Figure 1 and Table 1 below.

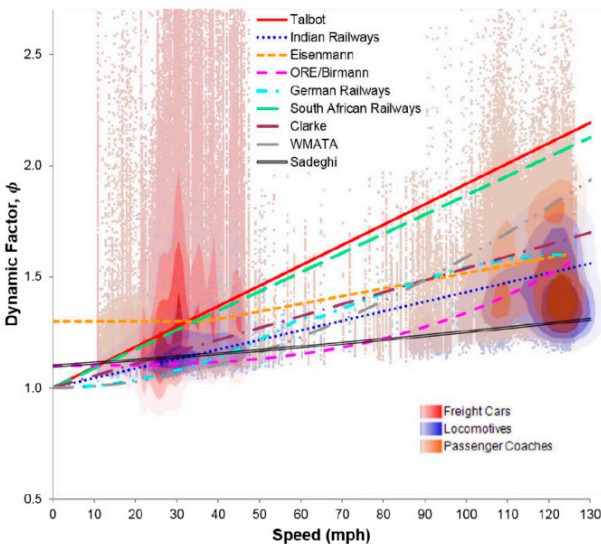

**Figure 1.** Different types of existing dynamic factor calculation methods [7].

**Table 1.** Parametric Comparison of Different Impact Factor Calculation methods [7].

| Evaluation Metric | Talbot | Indian Railways | Eisenmann | ORE/Birmann | German Railways | South African Railways | Clarke | WMATA | Sadeghi |
|---|---|---|---|---|---|---|---|---|---|
| Precent Exceeding | 0.23 | 0.61 | 0.37 | 0.75 | 0.56 | 0.25 | 0.45 | 0.48 | 0.89 |
| Mean Signed Difference $\sum \frac{(f(x_1)-y_i)}{n}$ | 0.20 | −0.19 | −0.081 | −0.25 | −0.16 | 0.16 | −0.10 | −0.074 | −0.31 |
| Mean percentage error $\frac{100\%}{n} \sum \frac{(f(x_i)-y_i)}{y_i}$ | 18 | −7.6 | 0.23 | −12 | −5.9 | 16 | −1.9 | −0.38 | −16 |
| Root Mean Square Deviation $\sqrt{\sum \frac{(f(x_i)-y_i)^2}{n}}$ | 0.61 | 0.53 | 0.51 | 0.57 | 0.56 | 0.59 | 0.52 | 0.57 | 0.57 |
| Speed − weighted signed difference $\frac{\sum (x_i f(x_i)-x_i y_i)}{\sum x_i}$ | 0.37 | −0.12 | −0.031 | −0.18 | −0.058 | 0.38 | −0.009 | 0.079 | −0.29 |
| Load − weighted signed difference $\frac{\sum (Q_i f(x_i)-Q_i y_i)}{\sum Q_i}$ | 0.24 | −0.13 | −0.018 | −0.19 | −0.11 | 0.20 | −0.051 | −0.027 | −0.25 |

Pita et al. provides a research regarding optimal track elasticity area in terms of track maintenance cost and vehicle energy consumption and as follows. As the track elasticity is abundant, the track maintenance cost decreases proportionally because the track load is relatively reduced. However, the energy consumption for train operation increases [8]. As such, the elastic change of the track shows opposite characteristic, and it is important to find the optimum point. Pita continues to claim that optimum track elasticity is proposed approximately 65~85 kN/mm [8]. Figure 1 shows the relation between track modulus and power dissipation in 300 km/h high-speed when a vehicle static load is 80 kN. This is because a light rail shows large displacement due to low stiffness which indicates low track modulus. In this regard, if the track modulus is increased by increasing track stiffness, that is, if the rail displacement decreased by increasing the track elastic modulus, it can lead to lower energy consumption. Bisadi et al. compares a dynamic factor calculation results using data where the vehicle is considered as a moving mass over simply supported girders using FEM analysis and compares the results with those of the standard design practice [9]. Van Dyk et al. outline many of the significant dynamic factor calculation methods applied in the different nations and their respective

evaluative metrics [7]. Kouroussis provides a review of the structural monitoring methods upon investigating the gauges and sensors used for field instrumentation, and thereby discusses some of the different parameters related to dynamic factor calculation methods [10]. Zhao et al. discuss a new method of determining the dynamic factor of railway tracks through the employment of multi-body dynamic model of vehicle and track interaction along with finite element modeling of wheel-rail contact to derive the low and high frequency dynamics [11]. Ding Youliang discusses the effects of lane position, number of train carriages, and speed of trains on dynamic factor by using the accumulative probability function of the Generalized Extreme Value Distribution, and subsequently proposes a probability distribution model for the dynamic factor [12]. In the current circumstances, the accuracy and methodology are discussed in great detail, but there are few studies concerning the applicability of a given conventional dynamic factor calculation method between different track structure types (ballasted and concrete tracks).

## 2.2. Application of Track Stiffness for Calculation of Dynamic Factor

When applied to ballasted and concrete tracks, the dynamic response of the wheel-rail contact load is different due to the respective track types' stiffness property [13,14]. In this regard, to derive the dynamic response of the track system, stiffness of the track should be a key factor. Prud'Homme's dynamic factor demonstrates this method, where the equation expresses the standard deviation of dynamic load as an equation related to train speed, variables according to rail and wheel defects, unsprung mass of the vehicle, track support stiffness, and damping of the track. The calculation method is as follows [15].

$$\sigma(\Delta QNS) = 0.45 \times \frac{V}{100} \times b \times \sqrt{m_{ns} \times K \times \gamma(\epsilon)} \tag{1}$$

where,

$\sigma(\Delta QNS)$: the standard deviation of DAF

$V$: train speed

$b$: variable of rail and wheel load contact point

$m_{ns}$: unsprung mass of the train

$K$: track support stiffness

$\gamma(\epsilon)$: rail damping

Among the variables, track support stiffness ($K$) is an indicator of the track structure's bearing capacity against dynamic loading of train wheel loads. This factor takes into consideration the structural characteristics and elastic range of the track system [15,16]. Track support stiffness can be understood as the vertical stiffness of the track structure, derived by the comprehensive spring coefficient from the individual components that comprise the entire track structure. The following equation is implemented where the spring coefficient $k_s$ is derived by calculating the spring coefficient (stiffness) of the individual components/layers of the ballasted track structure) and the spring coefficient $k_s$ is used to calculated the TSS of the structure [17]

$$k_{bs} = \frac{1}{\frac{1}{k_p} + \frac{1}{k_{sl}} + \frac{1}{k_b} + \frac{1}{k_{sg}}} \tag{2}$$

where,

$k_p$: the stiffness of the rail pad (kN/mm),

$k_{sl}$: the stiffness of the sleeper (due to the compressibility of wood in the rail-seat region and sleeper bending) (kN/mm),

$k_b$: the stiffness of the ballast layer (kN/mm),

$k_{sg}$: the stiffness of the subgrade (kN/mm), and

$k_{bs}$: the stiffness of the ballasted track structure (kN/mm)

$$k_{bss} = \frac{1}{\frac{1}{k_p} + \frac{1}{k_r} + \frac{1}{k_{sl}} + \frac{1}{k_{slab}} + \frac{1}{k_{sg}}} \tag{3}$$

where,

$k_p$: the stiffness of the pad (kN/mm),

$k_r$: the stiffness of the resilience (if applicable) (kN/mm),

$k_{sl}$: the stiffness of the sleeper (if applicable) (kN/mm),

$k_{slab}$: the stiffness of the ballast-less slab (kN/mm),

$k_{sg}$: the stiffness of the subgrade (kN/mm),

$k_{bss}$: the stiffness of the ballast-less slab track structure (kN/mm)

$$K_t = \sqrt[4]{\frac{64EI}{d^3}} K_s^3 \tag{4}$$

$E$: modulus of elasticity (GPa),

$I$: moment of inertia (mm$^4$),

$d$: distance of the sleepers (mm),

$k_s$: stiffness of the track structure (kN/mm),

$k_t$: track support stiffness (kN/mm).

For the trackside measurement of TSS, Hooke's law can be referred to in order to derive the track support stiffness of the track structure, whereby the track is considered as a beam structure, and the track support stiffness is derived using the variables of maximum dynamic wheel load during trackside measurement and the maximum rail displacement [18];

$$K_t = \frac{Q_{dyn}}{\delta_{max}} \tag{5}$$

where,

$K_t$: is the trackside measurement based track support stiffness (total spring constant) (kN/mm);

$P_{max}$: is dynamic wheel load (maximum value derived from trackside measurement) (kN);

$\delta_{max}$: is rail vertical displacement (maximum value derived from trackside measurement (mm).

The dynamic factor calculation method of KR C-14030 is used when designing the track. This is based on Eisenmann's equation that employs train vehicle speed and track quality as variables. However, according to the Prud'Homme equation, the dynamic coefficient is the square root of the track bearing stiffness [19]. Refer to Figure 2 on the correlation between Eisenmann and Prud'homme methods of dynamic factor calculation results with regards to track support stiffness and dynamic factor (track impact factor).

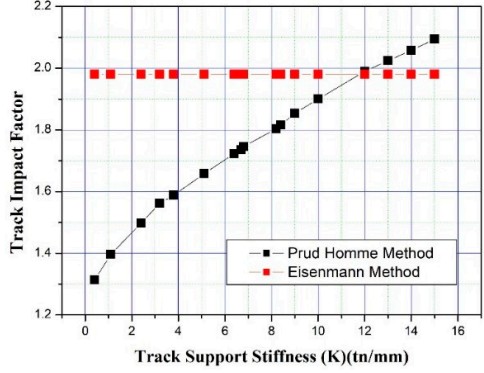

**Figure 2.** Correlation between Eisenmann and Prud'Homme method [19].

As a result of comparing the dynamic coefficients of Prud'Homme and Eisenmann, in the case of Eisenmann, the dynamic coefficient is constant regardless of the track support stiffness. In the case of the low track bearing stiffness section, the dynamic coefficient of the Eisenmann method is large, but in the track bearing stiffness of about 120 kN/mm or more, the dynamic coefficient of the Prud'Homme equation is larger than that of Eisenmann. The comparison results shown here prompts a requirement to investigate the application of dynamic factor evaluation of tracks based on the different stiffness properties of the tracks.

### 2.3. Dynamic Factor Calculation Methods Outlined

#### 2.3.1. Eisenmann Calculation Method ($DAF_{Eisenmann}$)

The dynamic factor method (as in accordance with the UIC method adopted in Korea) is currently used in both intercity and highspeed railway track design. Details of the Eisenmann's equation, variables, and parameters are shown in Table 2 (regarding exceeding probability application) and Table 3 (regarding track quality) explained below [20].

$$DAF = 1 + \phi$$
$$DAF = 1 + \phi\left(1.0 + 0.5\frac{V-60}{80}\right), \text{ in case of freight train}$$
$$DAF = 1 + \phi\left(1.0 + 0.5\frac{V-60}{190}\right), \text{ in case of passenger train} \tag{6}$$

**Table 2.** Track exceeding probability application condition [20–23].

| Exceeding Probability (%) | t | Application | Note |
|:---:|:---:|:---:|:---|
| 68.3 | 1 | Contact Stress, Track bed | t: Increase rate of standard deviation that depends on the confidence interval, value 3 is recommended for safety and reliability |
| 95.4 | 2 | Lateral load, Ballast | |
| 99.7 | 3 | Rail stress, Fastening systems, spring support (rail) | |

**Table 3.** Track quality and application condition.

| Track Quality | Φ | Note |
|:---:|:---:|:---|
| Very good | 0.1 | |
| Good | 0.2 | Φ: Coefficient dependent on track quality |
| Poor | 0.3 | |

Eisenmann's DAF (also represented as *i*) is calculated through the formula presented in the above Equation (6), and based on this, it is used by modifying it according to the characteristics of trains in Korea. In the case of the vehicle considering the wheel load caused by the track misalignment and cant shortage or cant excess, it is as follows.

$$Q_{eff} = Q \times 1.2 \tag{7}$$

Subsequently, the following formula is used for the dynamic load due to the exceptional impact caused by the vertical vibration of the vehicle's elastic part caused by irregularity at the wheel/rail contact point [9]

$$DAF_{Eisenmann} = Q_{eff} \times DAF \tag{8}$$

where,

$Q$: Static wheel load (kN),

$Q_{eff}$: Effective wheel load (kN),

$DAF_{Eisenmann}$: dynamic factor using the Eisenmann method

### 2.3.2. AREA Calculation Method (DAF$_{Area}$)

This calculation method is based on the geometrical characteristics of the wheel, which was proposed to increase by 1/100 of the value obtained by dividing 33 inches (83.8 cm) by the driving wheel diameter of the locomotive as the speed increases by 1 mile/hour. In the case of a diesel locomotive in operation, the wheel diameter is 40 inches (101.6cm), so 33/40 = 0.825 per mile, and when converted to km, it is 0.513%, as shown in the following equation [24].

$$DAF_{Area} = 1 + 0.513V/100 \tag{9}$$

where,
  $DAF_{Area}$: dynamic factor using the AREA method
  $V$: train speed

### 2.3.3. Trackside Measurement Based Calculation Method (DAF$_{Field}$)

Dynamic behavior analysis of track system, especially wheel load, should not only consider the static load but should consider the sum of static load and dynamic effect; the coefficient for the dynamic effect of the wheel load values obtained from field measurement data is used to calculate the wheel load fluctuation coefficient.

The reduction of the wheel weight occurs due to the vibration effect (e.g., hunting oscillation motion of the vehicle). At this time, the ratio of the static wheel load ($V_{static}$) to the wheel weight reduction value (dynamic load subtracted by static load) is called the wheel load fluctuation ratio. Wheel load fluctuation is calculated with the following equation:

$$Wheel\ load\ fluctuation\ coefficient = \frac{Dynamic\ Load\ (V_{Dynamic}) - Static\ Load\ (V_{static})}{V_{Static}} \tag{10}$$

Dynamic factor is influenced by the total spring constant of track, rail surface irregularities, and so on. The probability distribution of wheel load for dynamic factor analysis can be assumed as a normal distribution and shown in Figure 3 below:

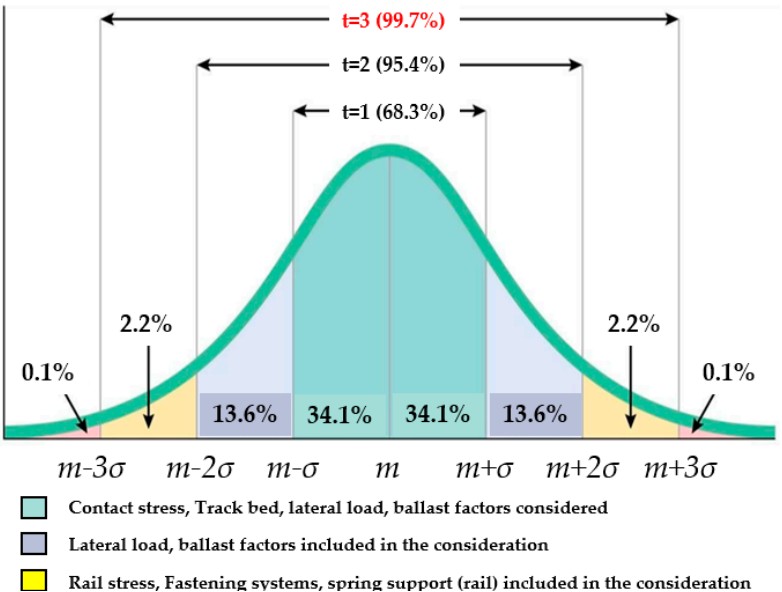

**Figure 3.** Wheel load fluctuation calculation method and parameter based on standard deviation.

Above normal distribution has maximum probability value at the mean (m) and descends as it moves away from the mean (m) and inflect x = m ± σ. In addition, as it moves away from the average,

the probability approaches 0, and the distribution curve and the area enclosed by the x-axis represent the probability. In this paper, the dynamic factor is calculated by considering the values of the range inclusive of t = 3 parameter ($m \pm 3\sigma$). Trackside measurement dynamic factor is calculated based on the standard deviation of dynamic wheel load (wheel load fluctuation), in which case can evaluate the level of the wheel-rail interaction force and the amplification level of the dynamic load acting on the track structure. The calculation method is as follows:

$$DAF_{Field} = 1 + at \tag{11}$$

where,

$a$: the slope of the regression line of the relation between the standard deviation of wheel load fluctuation and speed of the train vehicle

$t$: reliable area (within Figure 3) determining the inclusion range of the wheel load fluctuation coefficient standard deviation

## 3. Spring Constant of Track Types and Field Measurement for Dynamic Factor Comparison

According to Lee's study, as the track bearing stiffness increased for gravel and concrete roads, the dynamic coefficient also increased [3]. As a result of calculating the track bearing stiffness based on the measurement results, 130 to 160 kN on gravel roads /mm, it was found to have a value in the range of 25~65 kN/mm on concrete. The graph of the relationship between the track support stiffness and the dynamic coefficient of each track type according to his research results is shown in Figure 4.

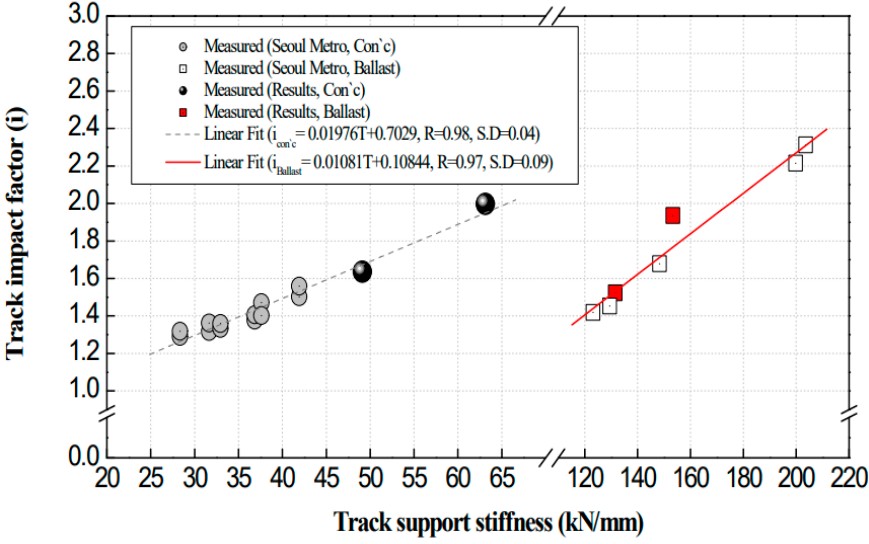

**Figure 4.** Correlation between track impact factor and track support stiffness [3].

As is mentioned in the introduction section, dynamic factor calculation methods vary between different nations and each method has been established based on numerous trial error and calculation efforts. Among these nations, Korea represents one of the many countries where the difference in the stiffness of the track properties is not properly considered, and dynamic factor evaluation between ballasted and concrete tracks structures is not properly conducted [24–26]. However, the above analysis indicates that dynamic factor calculation should at least consider the difference in the track spring constant or stiffness properties. In this regard, the difference of stiffness of the tracks in terms of the spring constant should be first outlined and be considered as a basis to evaluate the applicability of dynamic factor evaluation methods ($DAF_{Eisenmann}$ and $DAF_{Area}$).

*3.1. Trackside Measurement Conditions and Finite Element Analysis for Spring Constant Comparison*

3.1.1. Track Sites for Conducting Trackside Measurement and Finite Element Modeling Reference

In this study, the field measurement was performed on both ballasted track and concrete slab tracks found in the line of a newly constructed railway line (2018) [27]. Specification and overview of measurement sites are as shown in Table 4;

**Table 4.** Types of track evaluated.

| Track Types | Track Geometry (Lines) | Structure Type | Fastening System | Track Condition (Φ) |
|---|---|---|---|---|
| Ballasted track | Straight line | Earthwork Tunnel | E-clip | 0.3 |
| Concrete track | Straight line | Bridge Tunnel | System-300 | 0.3 |

3.1.2. Measuring Equipment

The main equipment used for field measurement are 1-axis, 2-axis strain gauge, and data acquisition system. Overview of field measurement system and specification of equipment are as shown in Table 5 below.

**Table 5.** Field instrumentation equipment for trackside measurement.

| Category | Type | Model | Manufacture Co. | Measurement Item |
|---|---|---|---|---|
| Sensor | 2 axis-strain gauge | FCA-5-11-1L | Tokyo Sokki | Vert./lateral Wheel load |
| | 1 axis-strain gauge | FLA-5-11-1L | Tokyo Sokki | Bending stress in rail |
| | Lateral Vertical Displacement Transducer (LVDT) | CDP-10 (10mm) | Tokyo Sokki | Vert./lateral dis. of rail, Vert. Dis. of sleeper |
| Measurement instrument | Data acquisition device for dynamic responses | MGC plus | HBM | |
| | | SDA-810 (8ch) | Tokyo Sokki | |
| | Bridge box | DB-120 (1ch, 8ch) | Kyowa | |
| | Laptop | Windows 10 | Samsung | |

3.1.3. Field Measurement Method

The wheel load was calculated by attaching a 2-axis strain gauge in eight directions with an angle of 45° to the neutral axis at a distance of 100 mm from the center of the sleepers. An example for strain gauge installation for wheel load measurement is shown in Figure 5. In the case of displacement, after fixing points were installed on the ballast track, the vertical displacement of the rail was measured. In the middle of the rail span between sleepers, the LVDT for the vertical rail displacement measurement of the rail was installed on the upper surface of the rail flange. An example for LVDT installation for vertical displacement measurement is shown in Figure 6. The elastic modulus of the rail, general value of 60kg/m rail steel ($E = 2.1 \times 10^5$ MPa), is applied. During the measurement, sufficient sampling rate (1 kHz) is set so as not to distort or cause the loss of data. In addition, in order to increase the reliability of the data, noise components other than wheel load are subjected to FFT (Fast Fourier Transform analysis) and conducted low-pass filtering and high-pass filtering.

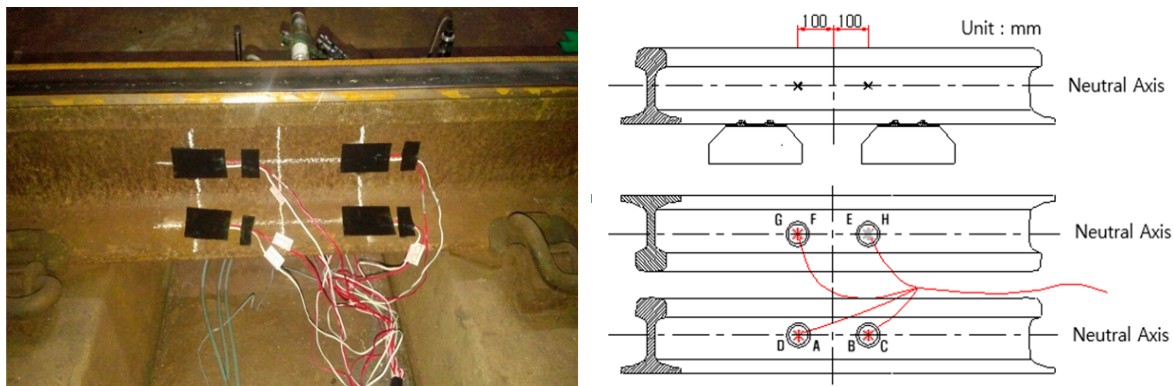

**Figure 5.** Strain gauge installation.

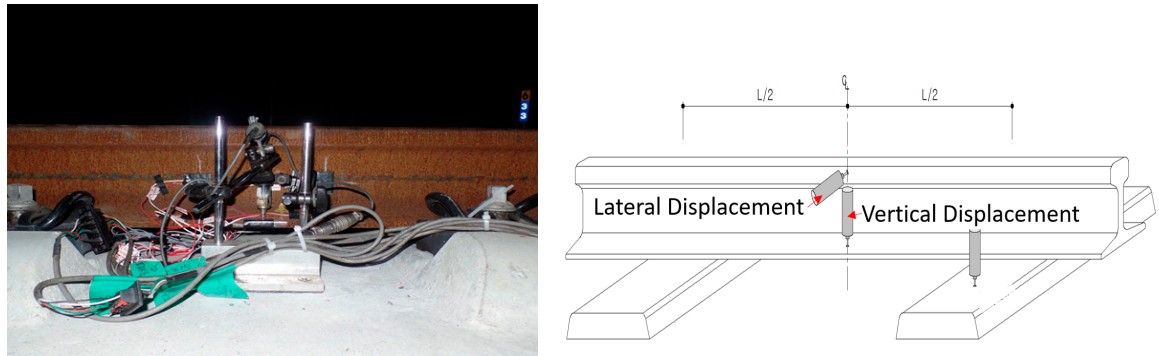

**Figure 6.** LVDT installation image.

Among the results, the measurement data for wheel load and vertical displacement is presented as an example in Figure 7.

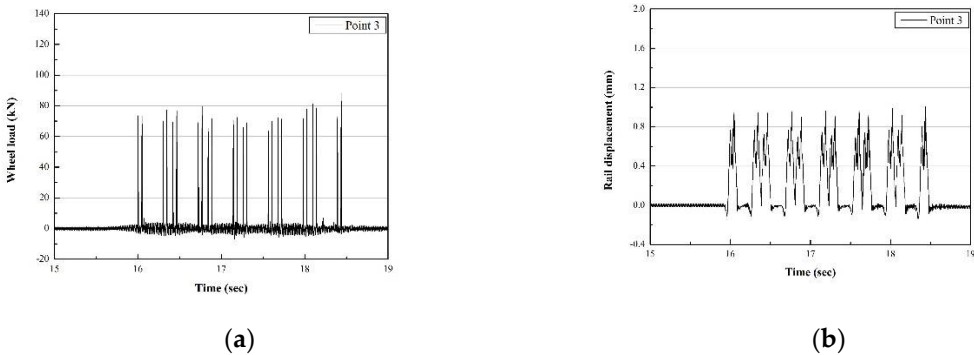

(**a**)                                              (**b**)

**Figure 7.** Data for the respective measurement factors (example), (**a**) wheel load; (**b**) vertical rail displacement.

### 3.2. Vehicle Specification

For the finite element method and the trackside measurement, the highspeed vehicle Korean Train Express (KTX) train (Static wheel load 85 kN) was used. Measurement results (the modeling conditions) were in accordance to train operation speed at the time, averaged (measurement results closest to the relevant speed) to 85, 100, 155, 185, 195, and 205. KTX train is comprised of 18 passenger trains, and the specification of the KTX are as shown in Table 6.

**Table 6.** Vehicle specification for trackside measurement.

| Division | Vehicle Specification (KTX) | Full Length (mm) | 2970 |
|---|---|---|---|
| Control method | VVVF(Variable Voltage Variable Frequency control) - IGBT(Insulated/Isolated Gate Bi-polar Transistor) | Max. speed (km/h) | 305 |
| Vehicle formation | PC1-T1-T2-T3-T4-T5-T6-T7-T8-PC2 PC: Power Car, T: Trailer car | Design max. speed (km/h) | 330 |
| Weight | Curb weight: 403ton, Full weight: 434ton | Gauge (mm) | 1435 |
| Whole length (m) | 201.00 | Design wheel load (kN) | 85 |

### 3.3. Finite Element Model Configuration

As the track sites selected for trackside measurement were based on Korean specification, the finite element modeling of standard rail structure of ballasted and concrete track structures were conducted in accordance to the KR-C 14030 design specification. In compliance to the design code, 3-dimensional solid element was applied to modeling rail and sleeper. Refer to Figure 8 and Table 7 for the details of finite element modeling.

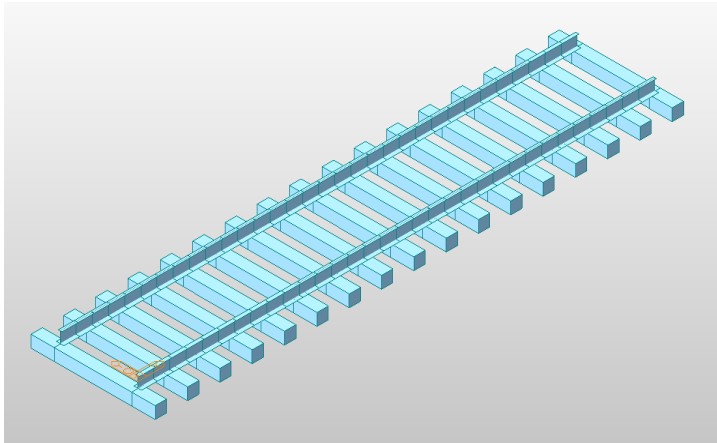

**Figure 8.** Dimensional analytical model.

**Table 7.** Track properties for finite element analysis.

| Division | Element | Item | Input Data |
|---|---|---|---|
| Rail | 3-dimensional solid element | Type | UIC 60 |
| | | Modulus of elasticity (kN/mm$^2$) | 206 |
| | | Weight density (kN/m$^2$) | 0.785 |
| | | Poisson's ratio | 0.30 |
| | | Height (mm) | 172 |
| | | Width of rail head (mm) | 72 |
| | | Width of rail base (mm) | 150 |
| | | Area (mm$^2$) | 76.86 |
| | | X axis moment of inertia (mm$^2$) | $3.05 \times 10^{-6}$ |
| | | Z axis moment of inertia (mm$^2$) | $5.13 \times 10^{-6}$ |
| | | Tensile strength (N/mm$^2$) | 880 |
| Rail pad | 3-dimensional spring-damper element | Height (mm) | 195 |
| | | Z axis moment of inertia (mm$^2$) | $1.41 \times 108$ |
| | | Vertical direction stiffness ($kw$) (kN/mm) | 100 |
| | | Vertical direction of damper coefficient, (kN•sec/mm) | 0.098 |

For the application of the train's dynamic load on the 3D model, the train load is idealized as a triangular form as shown in the Figure 5 below. Time history function analysis method is used,

in which the load applied at intervals of rate dependent of the applied speed. The load application is made to run at a constant speed according to the passage of time according to the axial arrangement of the KTX specification. The time difference between $t_1$ and $t_2$ is in Figure 9 is determined according to the train speed and the distance between nodes of the track structure model.

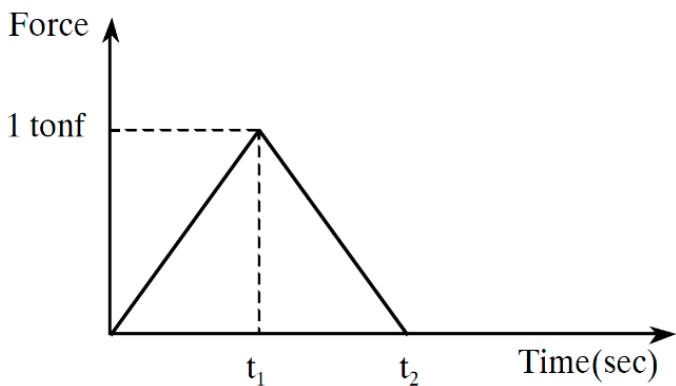

**Figure 9.** Dynamic load of the vehicle on the 3D model concept.

## 4. Results and Discussion

### 4.1. Finite Element Model Results

The finite model displacement results were derived by the dynamic loading conditions applied in the same conditions with field measurement site. Analysis result of vertical rail displacement according to train speed (85, 100, 155, 185, 195, 205 km/h) is shown in Table 8, where the displacement minimum and maximum displacement from the field measurement data are outlined.

**Table 8.** Field measurement result and finite element method analysis comparison.

| Item | | | Vertical Rail Displacement (mm) | | |
|---|---|---|---|---|---|
| **Track Types** | **Speed (km/h)** | **Finite Element Analysis** | **Field Measurement Data** | | |
| | | | **Min.** | **Max.** | |
| Ballasted | 85 | 0.697 | 0.37 | 1.05 | |
| | 95 | 0.706 | 0.43 | 1.09 | |
| | 155 | 0.749 | 0.41 | 1.07 | |
| | 185 | 0.760 | 0.57 | 1.11 | |
| | 195 | 0.767 | 0.62 | 1.18 | |
| | 205 | 0.772 | 0.51 | 1.13 | |
| Concrete | 85 | 1.154 | 0.42 | 0.98 | |
| | 95 | 1.162 | 0.49 | 1.01 | |
| | 155 | 1.215 | 0.58 | 1.28 | |
| | 185 | 1.238 | 0.62 | 1.24 | |
| | 195 | 1.247 | 0.71 | 1.12 | |
| | 205 | 1.256 | 0.83 | 1.08 | |

### 4.2. Field Measurement Results

In this paper, measuring equipment is installed to measure the track dynamic response according to train operation on ballasted and concrete track structures. The parameters required to measure the spring constant (load and displacement) and wheel load fluctuation (load and speed) are summarized below in Table 9. As can be seen in the results, the wheel load for the concrete and ballasted structure finite element model share the same load values, as they are both derived based on the Eisenmann theoretical model.

**Table 9.** Ballasted track field measurement result summary.

| Speed (km/h) | Wheel Load (kN) | | |
| --- | --- | --- | --- |
| | Concrete and Ballasted FEM (Eisenmann DAF × Static Load (85 kN)) | Ballasted Track Field Measurement | Concrete Track Field Measurement |
| 85 | 139.32 | 94.21 | 89.34 |
| 95 | 141.36 | 101.9 | 87.41 |
| 155 | 148.75 | 97.89 | 88.24 |
| 185 | 152.75 | 125.3 | 95.82 |
| 195 | 154.11 | 124.3 | 98.1 |
| 205 | 155.47 | 123.2 | 101.3 |

### 4.3. Total Spring Constant of the FEM Models and Track Calculation Result

The total spring constant of track according to finite element analysis result is calculated and compared with the value derived from field measurement (shown in Tables 10 and 11). To calculate total spring constant of track in ballasted track, Equation (5) was used. In addition, using wheel load and vertical rail displacement from field measurement, total spring constant of track is calculated in both ballasted track and concrete tracks (as is shown from Tables 12 and 13 below). The total spring constant of the respective track types, both 3D models and the real track sites, are outlined in a graph format in Figure 10.

**Table 10.** Total spring constant of ballasted track (FEM).

| Train Speed (km/h) | Wheel Load (kN) | Vertical Rail Displacement (mm) | Total Spring Constant of Track (kN/mm) |
| --- | --- | --- | --- |
| 85 | 139.32 | 0.697 | 199.88 |
| 95 | 141.36 | 0.706 | 200.23 |
| 155 | 148.75 | 0.749 | 198.59 |
| 185 | 152.75 | 0.760 | 200.98 |
| 195 | 154.11 | 0.767 | 200.93 |
| 205 | 155.47 | 0.772 | 204.38 |

**Table 11.** Total spring constant of concrete track (FEM).

| Train Speed (km/h) | Wheel Load (kN) | Vertical Rail Displacement (mm) | Total Spring Constant of Track (kN/mm) |
| --- | --- | --- | --- |
| 85 | 139.32 | 1.154 | 120.72 |
| 95 | 141.36 | 1.162 | 121.65 |
| 155 | 148.75 | 1.215 | 122.42 |
| 185 | 152.75 | 1.238 | 123.38 |
| 195 | 154.11 | 1.247 | 123.58 |
| 205 | 155.47 | 1.256 | 123.78 |

**Table 12.** Total spring constant of ballasted track (Field measurement).

| Train Speed (km/h) | Wheel Load (kN) | Vertical Rail Displacement (mm) | Total Spring Constant of Track (kN/mm) |
| --- | --- | --- | --- |
| 85 | 94.21 | 1.05 | 89.72 |
| 95 | 101.9 | 1.09 | 93.48 |
| 155 | 97.89 | 1.07 | 91.48 |
| 185 | 125.3 | 1.11 | 112.88 |
| 195 | 124.3 | 1.18 | 105.33 |
| 205 | 123.2 | 1.13 | 109.02 |

**Table 13.** Total spring constant of concrete track (Field measurement).

| Train Speed (km/h) | Wheel Load (kN) | Vertical Rail Displacement (mm) | Total Spring Constant of Track (kN/mm) |
|---|---|---|---|
| 85 | 89.34 | 0.98 | 91.16 |
| 95 | 87.41 | 1.01 | 86.54 |
| 155 | 88.24 | 1.28 | 68.93 |
| 185 | 95.82 | 1.24 | 77.27 |
| 195 | 98.1 | 1.12 | 87.58 |
| 205 | 101.3 | 1.08 | 93.79 |

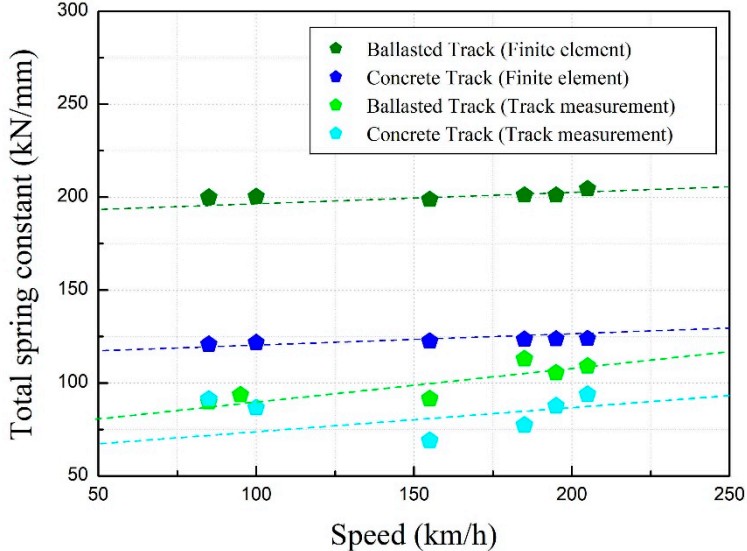

**Figure 10.** Total spring constant comparison.

Total spring constant comparison results showed the following results. Between the finite element tracks, concrete track total spring constant is about 61% of the ballasted track. Between the real track sites, the concrete track total spring constant is about 85% of the ballasted track. Between the finite element track to the real track site comparisons, the actual ballasted track is 50% compared to the 3D model version, and for the 68% concrete track site comparison. As a result of overall comparison analysis, it is judged that the currently applied dynamic factor application method will bring about a deviation between the track types. This indicates that the dynamic factor calculation methods should be reconsidered to derive a separate application method that considers each track type's stiffness properties due to the evident differences.

## 5. Dynamic Factor Calculation Result

For the calculation of the dynamic factors of the trains in accordance to the difference speed parameters, the standard deviation of the wheel weight fluctuation is considered to be three times the value within the range of 99.7% (excess probability inclusion rate up to t = 3, refer to Figure 3), as is complaint to the standard dynamic load calculation method outlined in KR C-14030. Wheel load fluctuation standard deviation calculation is shown in Table 14.

Based on the standard deviation value, $DAF_{Field}$ was calculated accordingly for the respective track types. The result is summarized in Figure 11.

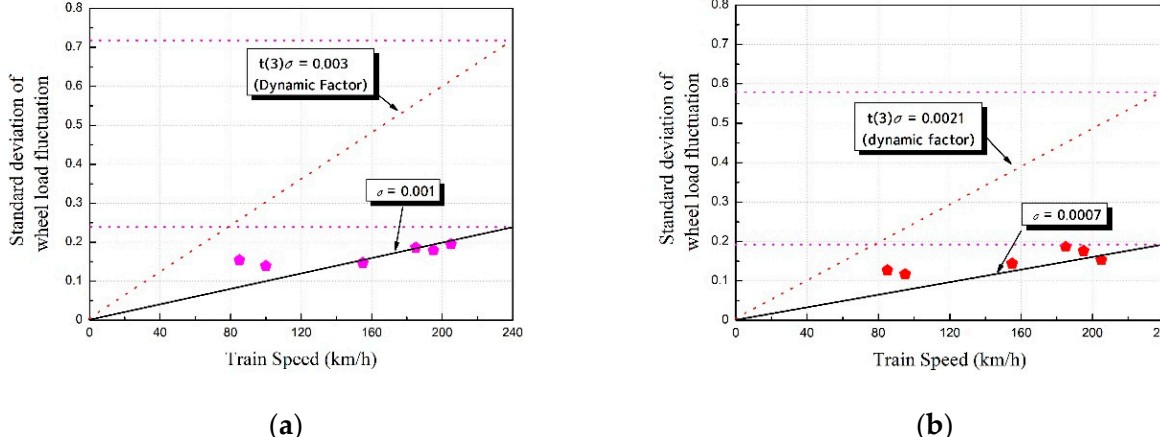

**Figure 11.** Standard deviation of wheel load fluctuation comparison, (**a**) ballasted track DAF$_{\text{Field}}$ result. (**b**) concrete track DAF$_{\text{Field}}$ result.

**Table 14.** Dynamic Stability Evaluation Parameter Comparison of Ballasted and Ballast-less tracks.

| Track Structure | Train Speed (km/h) | Static Wheel Load (kN) | Average Wheel Load Fluctuation | Standard Deviation of Wheel Load Fluctuation |
|---|---|---|---|---|
| Ballasted track | 85 | | 1.15 | 0.131 |
| | 100 | | 1.17 | 0.139 |
| | 155 | | 1.33 | 0.147 |
| | 185 | | 1.47 | 0.186 |
| | 195 | | 1.53 | 0.179 |
| | 205 | 85 | 1.52 | 0.195 |
| Concrete track | 85 | | 1.13 | 0.127 |
| | 100 | | 1.24 | 0.117 |
| | 155 | | 1.26 | 0.144 |
| | 185 | | 1.24 | 0.186 |
| | 195 | | 1.33 | 0.176 |
| | 205 | | 1.38 | 0.153 |

Based on the calculations of the each of the DAF conditions (DAF$_{\text{Area}}$, DAF$_{\text{Eisenmann}}$, and DAF$_{\text{Field}}$), a comprehensive comparison was conducted in accordance with the train operation speed for ballasted and concrete tracks. Results are summarized in Table 15 and Figure 12 below.

**Table 15.** Dynamic Factor Comparison of Ballasted and Concrete tracks.

| Track Structure | Speed (km/h) | Dynamic Factor | | |
|---|---|---|---|---|
| | | DAF$_{\text{Area}}$ | DAF$_{\text{Eisenmann}}$ | DAF$_{\text{Field}}$ |
| Ballasted | 85 | 1.436 | 1.639 | 1.229 |
| | 100 | 1.513 | 1.663 | 1.270 |
| | 155 | 1.795 | 1.750 | 1.419 |
| | 185 | 1.949 | 1.797 | 1.499 |
| | 195 | 2.000 | 1.813 | 1.527 |
| | 205 | 2.052 | 1.829 | 1.554 |
| Concrete track | 85 | 1.436 | 1.639 | 1.263 |
| | 100 | 1.513 | 1.663 | 1.347 |
| | 155 | 1.795 | 1.750 | 1.368 |
| | 185 | 1.949 | 1.797 | 1.389 |
| | 195 | 2.000 | 1.813 | 1.410 |
| | 205 | 2.052 | 1.829 | 1.473 |

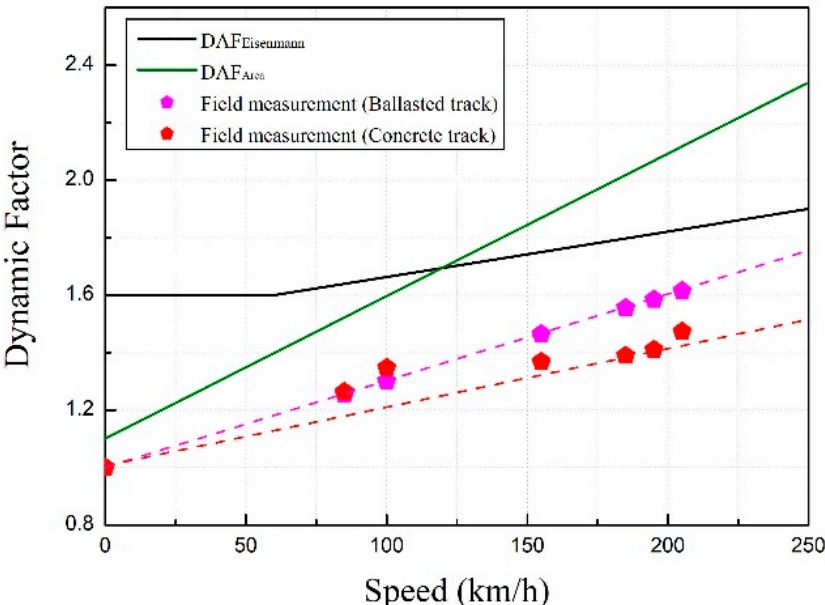

**Figure 12.** Comprehensive dynamic factor comparison.

## 6. Result and Conclusions

The comparison results of $DAF_{Eisenmann}$, $DAF_{Area}$, and $DAF_{Field}$ is as follows. As the variables for the $DAF_{Eisenmann}$ and $DAF_{Area}$ calculation for both ballasted and concrete track structures are the same (using Equations (8) and (9), respectively), their results are obviously the same when compared with one another for each train speed conditions. Between the two dynamic factor calculation types, however, slope of the $DAF_{Area}$ is visibly higher than that of the $DAF_{Eisenmann}$ (this result is again expected as it has been established since the research results outlined in Figure 1). However, the difference between the $DAF_{Field}$ between the concrete and ballasted tracks were relatively more noticeable. Throughout the speed variables, the comparison of $DAF_{Field}$ of ballasted tracks to the $DAF_{Eisenmann}$ and $DAF_{Area}$ showed a range from 17% up to 32%, and 18% to 34% respectively. In the case of the $DAF_{Field}$ of concrete track, again the difference was higher, showing a range of from 14% to 42% for the $DAF_{Eisenmann}$ and 23% to 30% for the $DAF_{Area}$. Between the track types of the $DAF_{Field}$, a difference from 4% to 8% was observed. Results of this study proposed and demonstrated that dynamic factor calculations based on the standard methods deviate at a relatively large degree from the track results, and there is a difference in the results between the track types. In theory, the dynamic factor should be a reliable indicator of a track structure's dynamic response performance. However, the lack of consistency and the deviation from the standard dynamic factor calculation between the track types shows that there is bound to be a margin of error in the dynamic factor evaluation of tracks if the current standard codes are continued to be used without distinction between the track classes.

**Author Contributions:** Conceptualization, J.L., and K.O.; methodology, K.O.; validation, Y.P. and J.C.; formal analysis, J.L.; investigation, J.L. and K.O.; data curation, J.L.; writing—original draft preparation, J.L. and K.O.; writing—review and editing, J.L. and K.O.; supervision, J.C. and Y.P.; project administration, J.C. All authors have read and agreed to the published version of the manuscript.

**Funding:** This work is supported by the Korea Agency for Infrastructure Technology Advancement (KAIA) grant funded by the Ministry of Land, Infrastructure and Transport (Grant 20CTAP-C152026-02). Also, this work was supported by 2020 BUCHEON UNIVERSITY Research Grant.

**Conflicts of Interest:** The authors declare no conflict of interest.

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
