# Peer review of "Study on the Applicability of Dynamic Factor Standards by Comparison of Spring Constant Based Dynamic Factor of Ballasted and Concrete Track Structures"

_applsci, doi:10.3390/app10238361_

Round 1

Reviewer 1 Report

Reviewed paper is related to dynamic factor evaluation based on total spring constant of track. Authors present numerical results from FEM and experimental results from field measurements. In my opinion topic is interesting, abstract clearly underline achieved results. State of the art could be improved in my opinion. Bellow I have listed my comments:

  1. First of all: Authors need to carefully read and edit especially taking into account the figure numbers in references.
  2. Results obtained by FEM method gives relatively large error in comparison to field measurement results. Why authors not improved model? What it the purpose to use FEM model without agreement with experimental results?
  3. FEM model: why such a length (10m)? Why such discretisation, did authors utilised other mesh densities?
  4. I suggest to present some example of results from measurements in the form of signals from strain gauges/displacement sensors. It will cause that the paper will more interesting.
  5. Results with table 12 does not correlate with results in fig. 7 (e.g. total spring for 85 km/h – 199.88 in table @ ballasted and in fig 7 much smaller value).
  6. 7: FEM results (green) are for ballasted case? Ballasted and ballasted-less results (red and magenta) come from field measurements? It need to be explained, clearly marked in figure.
  7. 2: low resolution/image quality
  8. 3 - „Fig. 1.3 shows…”. There is lack of fig. 1.3.
  9. 6, row 179: “and Fig. It has the same graph shape as 1 [7].”. Lack of fig. number. What means “as 1”?
  10. 7, row 202: KTX – please explain what means this abbreviation
  11. 9: LVDTs – please explain abbreviation
  12. 5: Caption of figure indicates that only strain gauge is utilised but this figure illustrates also location of displacement sensors. Please change the caption
  13. 9, rows 238-240: “…roads. Rail stress) The dynamic…” – something is wrong with the sentence
  14. 9: “Tables 7 and 8 below summarize and summarize”. Please remove one “summarize”
  15. 15, row 331: “Fig.” – lack of figure number. Next sentence “It is shown in 6, and is as follows.”. What is the number 6 related to?
  16. Row 339: “track support of each track structure It is considered appropriate” – sentence was not finished with “.”
  17. Row 346: “The evaluation results are as follows;” please use “:” instead “;” in whole paper

Author Response

Reviewer 1

The authors of the article would like to extend their utmost appreciation the reviewer for their time and effort spent to improve the quality of the paper.

The authors have prepared the following responses to the comments and points of revision. We have first provided the quote on the specific point specified the reviewer that the authors intend to address, and provided the relevant response.  Please refer to the below sections for details;

Reviewed paper is related to dynamic factor evaluation based on total spring constant of track. Authors present numerical results from FEM and experimental results from field measurements. In my opinion topic is interesting, abstract clearly underline achieved results. State of the art could be improved in my opinion. Bellow I have listed my comments

1.First of all: Authors need to carefully read and edit especially taking into account the figure numbers in references.

Response:

The figure and table numbering, has been revised throughout the paper. Please refer to the revised version of the paper for details

2. Results obtained by FEM method gives relatively large error in comparison to field measurement results. Why authors not improved model? What it the purpose to use FEM model without agreement with experimental results?

Response:

The purpose is to precisely illustrate that the FEM model does not agree with the experimental results. Even though the FEM model is conducted in compliance to the standard code in Korea (and agrees with the UIC code), there is a significant deviation from the field measurement. However, deviation of the results of the measurement parameters (wheel load and displacement) is usually expected, as of course in the field measurement, there are environmental factors that cannot be accounted for the FEM modelling. The problem however, is the subsequent dynamic factor evaluation, where based on which standard is used, the deviation becomes even greater. And this is especially problematic between concrete and ballasted type track structures. This explanation has bene made more clear in the revised version of the paper (please refer to lines 65 to 78, and the revised section 2).

3. FEM model: why such a length (10m)? Why such discretisation, did authors utilised other mesh densities?

Response:

The FEM model was conducted based on the modelling method compliant to the KRC-14030 code. Please refer to the revised version lines 305 ~321 for the added details regarding this matter.

4.I suggest to present some example of results from measurements in the form of signals from strain gauges/displacement sensors. It will cause that the paper will more interesting.

Response:

Example results for the measurements have been included in the revised version’s Figure 7 (only examples have been included as there were quite a number of datas that were used. Thank you kindly for the advice.

5. Results with table 12 does not correlate with results in fig. 7 (e.g. total spring for 85 km/h – 199.88 in table @ ballasted and in fig 7 much smaller value).

Response:

Results have been revised throughout the paper to be consistent. Please refer to the revised paper results section for details.

6. FEM results (green) are for ballasted case? Ballasted and ballasted-less results (red and magenta) come from field measurements? It need to be explained, clearly marked in figure.

7. Figure 2: low resolution/image quality

Response:

Figure 2 resolution has been fixed

8. Fig. 1.3 shows…”. There is lack of fig. 1.3.

Response:

The line has been revised

9. row 179: “and Fig. It has the same graph shape as 1 [7].”. Lack of fig. number. What means “as 1”?

Response:

The line has been revised

10. row 202: KTX – please explain what means this abbreviation

Response:

Explanation of the abbreviation has been added (line 298 to 299)

11. LVDTs – please explain abbreviation

Response:

Explanation of the abbreviation has been added (please refer to the revised version Table 5)

12. Caption of figure indicates that only strain gauge is utilised but this figure illustrates also location of displacement sensors. Please change the caption

Response:

Caption for Figure 6 has been revised

13. Rows 238-240: “…roads. Rail stress) The dynamic…” – something is wrong with the sentence

Response:

The line has been revised

14. “Tables 7 and 8 below summarize and summarize”. Please remove one “summarize”

Response:

Summarize has been removed

15. row 331: “Fig.” – lack of figure number. Next sentence “It is shown in 6, and is as follows.”. What is the number 6 related to?

Response:

The line has been revised

16. Row 339: “track support of each track structure It is considered appropriate” – sentence was not finished with “.”

Response:

The line has been revised

17. Row 346: “The evaluation results are as follows;” please use “:” instead “;” in whole paper

Response:

Usage of semicolon has been  replaced with “:” throughout the paper in the appropriate sections.

The authors would like to express their sincerest gratitude for the reviewers’ efforts and time taken to review and comment on this article. Thanks to your esteemed feedbacks, the authors believe the paper was able to undergo a significant improvement since the previous version. The authors hope that the revised version satisfies the points of inquiries provided by the reviewers.

Thank you

Reviewer 2 Report

Review for Manuscript ID applsci-986842

Evaluation of Dynamic Factor Considering Total Spring Constant of Track of Ballasted and Ballast-less Tracks

by Yonggul Park *, Kyuhwan Oh, Jaeik Lee.

The manuscript is talking on the dynamic factor of a railway track. In the present state the manuscript cannot be published and not reviewed. It is confusing. Formula and symbols are not unique. Sentences are in contradiction. Calculated dynamic factors are presented, but not a corresponding model. Results are shown as data points. No time history, no spectrum is given to judge about the quality of the results. After the quality of the manuscript has been improved, a review can be made.

Author Response

Reviewer 2

The authors of the article would like to extend their utmost appreciation the reviewer for their time and effort spent to improve the quality of the paper.

The authors have prepared the following responses to the comments and points of revision. We have first provided the quote on the specific point specified the reviewer that the authors intend to address, and provided the relevant response.  Please refer to the below sections for details;

The manuscript is talking on the dynamic factor of a railway track. In the present state the manuscript cannot be published and not reviewed. It is confusing. Formula and symbols are not unique. Sentences are in contradiction. Calculated dynamic factors are presented, but not a corresponding model. Results are shown as data points. No time history, no spectrum is given to judge about the quality of the results. After the quality of the manuscript has been improved, a review can be made

Response,

To give a brief summary of the purpose of the paper:

Dynamic factors are comparable between different track structures, as dynamic factor is a coefficient of that is used to essentially determine how much of the dynamic load is exceeding the design specification based static load. The methods described in Figure 1 are the dynamic factor calculation methods used all over the world. The purpose is to illustrate that the standard calculation of the dynamic factor does not agree with the experimental results. As is shown through the modelling,eEven though the FEM model is conducted in compliance to the standard code in Korea (and agrees with the UIC code), there is a significant deviation from the field measurement. However, deviation of the results of the measurement parameters (wheel load and displacement) is usually expected, as of course in the field measurement, there are environmental factors that cannot be accounted for the FEM modelling. The problem however, is the subsequent dynamic factor evaluation, where based on which standard is used, the deviation becomes even greater. And this is especially problematic between concrete and ballasted type track structures. This explanation has bene made more clear in the revised version of the paper (please refer to lines 65 to 78, and the revised section 2).

Furthermore, details of the finite element modelling method and results has been included, including the dynamic load histogram and conditions.

The authors would like to express their sincerest gratitude for the reviewers’ efforts and time taken to review and comment on this article. Thanks to your esteemed feedbacks, the authors believe the paper was able to undergo a significant improvement since the previous version. The authors hope that the revised version satisfies the points of inquiries provided by the reviewers.

Thank you

Reviewer 3 Report

Author presented good research and their results are very interesting in the railway transport. Expecially for problematic of construction and maintenance of track lines.

In the fugure 1. authors compare different values of "Dynamic factor". But I have question: are these values comparable? Because authors compare different coutry and thesy have different railway guage (f.e. in the India is 1,676 mm railway guage and in the Germany is standart rail guage 1,435 mm).

I recommend to authors improve the figure 2. It has bad quality (higher density of image is needed).

Author Response

Reviewer 3

The authors of the article would like to extend their utmost appreciation the reviewer for their time and effort spent to improve the quality of the paper.

The authors have prepared the following responses to the comments and points of revision. We have first provided the quote on the specific point specified the reviewer that the authors intend to address, and provided the relevant response.  Please refer to the below sections for details;

Comment 1

In the fugure 1. authors compare different values of "Dynamic factor". But I have question: are these values comparable? Because authors compare different coutry and thesy have different railway guage (f.e. in the India is 1,676 mm railway guage and in the Germany is standart rail guage 1,435 mm).

Dynamic factors are comparable between different track structures, as dynamic factor is a coefficient of that is used to essentially determine how much of the dynamic load is exceeding the design specification based static load. The methods described in Figure 1 are the dynamic factor calculation methods used all over the world. The purpose is to illustrate that the standard calculation of the dynamic factor does not agree with the experimental results. As is shown through the modelling,eEven though the FEM model is conducted in compliance to the standard code in Korea (and agrees with the UIC code), there is a significant deviation from the field measurement. However, deviation of the results of the measurement parameters (wheel load and displacement) is usually expected, as of course in the field measurement, there are environmental factors that cannot be accounted for the FEM modelling. The problem however, is the subsequent dynamic factor evaluation, where based on which standard is used, the deviation becomes even greater. And this is especially problematic between concrete and ballasted type track structures. This explanation has bene made more clear in the revised version of the paper (please refer to lines 65 to 78, and the revised section 2).

Comment 2

I recommend to authors improve the figure 2. It has bad quality (higher density of image is needed).

Figure 2 resolution has been fixed

The authors would like to express their sincerest gratitude for the reviewers’ efforts and time taken to review and comment on this article. Thanks to your esteemed feedbacks, the authors believe the paper was able to undergo a significant improvement since the previous version. The authors hope that the revised version satisfies the points of inquiries provided by the reviewers.

Thank you